# Hepatoprotective Effects of Rosmarinic Acid on Ovalbumin-Induced Intestinal Food Allergy Mouse Model

**DOI:** 10.3390/molecules28020788

**Published:** 2023-01-12

**Authors:** Binmei Jia, Jieli Shang, Haolong Zeng, Xuanpei Wang, Min Fang, Lin Xu, Xin Liu, Kejia Wu, Zhiyong Gong, Qing Yang

**Affiliations:** 1Key Laboratory for Deep Processing of Major Grain and Oil, Ministry of Education, Wuhan Polytechnic University, Wuhan 430023, China; 2Food Safety Research Center, Key Research Institute of Humanities and Social of Hubei Province, Wuhan 430023, China; 3Department of Laboratory Medicine, Tongji Hospital, Tongji Medical College, Huazhong University of Science and Technology, Wuhan 430030, China

**Keywords:** rosmarinic acid, anti-allergic, hepatoprotective effects, anti-oxidative, anti-inflammatory

## Abstract

Rosmarinic acid (RA) has been proven to exert antianaphylaxis in atopic dermatitis, asthma, and allergic rhinitis. The aim of this study was to determine the hepatoprotective effects of RA on ovalbumin (OVA) challenge-induced intestinal allergy. The results exhibited that RA could relieve anaphylactic symptoms, decrease diarrhea, and prevent hypothermia in allergic mice. Moreover, the elevation of OVA specific IgE (OVA-sIgE), histamine, and mouse mast cell proteinases (mMCP-1) in the serum of OVA challenged mice were remarkably inhibited by RA. OVA challenge resulted in notable increases in serum alanine aminotransferase (ALT), aspartate aminotransferase (AST) activities, liver malondialdehyde (MDA) and nitic oxide (NO) levels, and a remarkable decrease in liver superoxide dismutase (SOD) activity and glutathione (GSH) level. RA treatments succeeded in improving these biochemical parameters and promote the redox homeostasis. Cytokine expression evaluation showed that RA effectively enhanced the expression of anti-inflammatory cytokines (IL-10 and FOXP-3) in the liver of OVA-challenged mice. Meanwhile, the elevation of pro-inflammatory cytokines (TNF-α, IL-4, IL-6, mMCP-1, and iNOS) were remarkably inhibited by RA. These findings suggest that RA possesses hepatoprotective effects on OVA challenge-induced liver injury. The anti-oxidative and anti-inflammatory activities of RA potentially play vital roles in this process.

## 1. Introduction

Food allergy (FA) is an adverse immune reaction to certain foods containing allergens. The adverse health effects of food allergy involve a serious of symptoms ranging from mild irritations to life-threatening anaphylaxis, which may affect 2–10% of the population worldwide [1]. Due to the increasing prevalence and life-threatening potential, it has become one of the growing health problems around the world [2]. Among the “big eight” allergic foods, hypersensitivity caused by *Gallus gallus* egg is a common situation ordinarily affecting up to 9% of children around the world [3]. In China, 3.9% of the rural population is sensitized to egg [4]. The OVA, as a major allergen of egg [5], has been commonly used in establishing a food allergy mouse model for research on the pathogenesis, prevention, and treatment of food allergy [6].

In food allergy, the increase in intestinal mucosal permeability leads to the entrance of food antigens to the liver through portal vein circulation. Generally, the liver, which acts as an immune tolerant organ, is tolerant to food antigens. However, prolonged exposure to antigens leads to the imbalance of the gut–liver axis, which in turn activates the liver immune response and ultimately causes liver damage [7,8,9]. Oxidative stress plays an indispensable part in food allergy-induced liver damage accompanied by the reduction in SOD activity, dropped GSH level, and the increasing levels of malondialdehyde (MDA) and nitric oxide (NO) [10]. 

Nowadays, the administration or intake of natural bioactive compounds has become a key factor to improve the health status and deal with liver dysfunction from a variety of causes. For instance, in metabolic dysfunction-associated fatty liver disease (MAFLD), various bioactive ingredients (e.g., vitamin D, protein hydrolysates from anchovy, and parthenolide) have shown potential health benefits [11,12,13]. Natural antioxidants such as polyphenols are well-known to possess hepatoprotective activity based on its antioxidant properties, which promote the redox homeostasis of liver [14]. In addition, regulation of liver inflammation through restoring the balance between pro-inflammatory and anti-inflammatory cytokine expression is also an important way for these antioxidants to exert hepatoprotective activity [15]. 

RA, an ester of caffeic acid and 3, 4-dihydroxyphenyl lactic acid, is an active polyphenol found in medicinal herbs including *Ocimum basilicum*, *Salvia miltiorrhiza Bunge*, *Rosmarinus officinalis*, and *Origanum vulgare* [16]. Given its strong antioxidative properties and immune-modulating ability, RA has been widely known for a wide range of pharmacological activities such as anti-inflammatory, hepatoprotective activity, cardioprotective, and anti-allergic activity [17,18,19]. For anti-anaphylaxis, RA has been widely reported to possess protective effects on atopic dermatitis, asthma, and allergic rhinitis [18]. However, little is known about the health effect of RA on intestinal food allergy. 

Given the anti-allergic and hepatoprotective activity of RA, we were encouraged to study the health effects of RA on liver damage caused by OVA oral challenge. Therefore, the effects of RA on redox state, activity of AST and ALT in the serum, gene expression of inflammatory cytokines as well as pathological changes of liver were assayed to evaluate its hepatoprotective activity on liver damage in food allergy.

## 2. Results

### 2.1. Attenuation of Anaphylaxis by Rosmarinic Acid on OVA-Induced Allergic Mice

To assess the anti-allergic effect of RA, the animal experiment was performed according to the animal experiment procedure shown in Figure 1A. OVA-sensitized mice were once daily administrated with RA (30 mg.kg^−1^, RA-Low; 90 mg.kg^−1^, RA-Middle; 270 mg.kg^−1^, RA-High) by gavage during the OVA challenge phase. The anaphylactic score of allergy symptoms assigned to different groups of mice were conducted according to the allergy score standard listed in Table 1. As shown in Figure 1B,C, the anaphylactic scores and diarrhea rates of the mice in the model group were remarkably (*p* < 0.01) higher than the mice in the control group, indicating the success of the modeling. Continuous administration with different doses of RA for 14 days significantly suppressed the allergic symptoms of mice with decreased anaphylactic scores (Figure 1B) and dropped the diarrhea rates (Figure 1C). Moreover, the rectal temperature of the mice in the OVA challenged group decreased by ~2.23 °C at 45 min after the last challenge whereas the rectal temperatures were only reduced by ~2.0 °C, ~0.9 °C and ~0.3 °C of the mice treated by 30, 90, and 270 mg.kg^−1^ RA, respectively (Figure 1D). This indicates that hypothermia induced by OVA challenge was remitted by the administration of RA. These results prove that RA significantly alleviated the anaphylactic symptoms in OVA-challenged mice.

### 2.2. Regulating Effects of RA on Antibodies and Anaphylactic Mediators

After the fifth OVA oral challenge, sera were collected to determine the allergic conditions. As shown in Figure 2A, the concentration of OVA-sIgE in the sera of OVA challenged mice was up to 19,996.8 ng/mL, while the concentration was decreased to 10,517.0 ng/mL in mice treated with 270 mg/kg RA. Low and middle dose of RA also showed an inhibition effect, although there was no significant difference. Additionally, the concentration levels of histamine and mMCP-1 in the serum were analyzed as an evaluation of mast cell activation and degranulation. The results showed that the histamine and mMCP-1 levels were both remarkably (*p* < 0.01) increased in the OVA-challenged mice (histamine: 980.9 ng/mL; mMCP-1: 1277.5 ng/mL) compared to the negative control (histamine: 227.0 ng/mL; mMCP-1: 51.4 ng/mL) (Figure 2B,C). The mice administrated with RA showed lower histamine levels in all groups (RA-Low: 884.6 ng/mL; RA-Middle: 816.4 ng/mL; RA-High: 802.7 ng/mL) (Figure 2B). A significant (*p* < 0.01) reduction in the histamine levels was observed in the high-dose of RA treated group. In all of the RA treated mice, the mMCP-1 concentrations showed lower levels compared with the OVA-allergic group, while remarkable (*p* < 0.01) reductions were observed in the middle and high dose of RA treatment (RA-Middle: 653.4 ng/mL; RA-High: 566.0 ng/mL) (Figure 2C). The above results provide further evidence that RA might promote the oral tolerance to OVA allergen in mice.

### 2.3. Effects of RA on Liver Tissue Morphology and Serum ALT and AST

The protection effects of RA against liver dysregulation in food allergic mice were evaluated by histopathological observation and the detection of transaminase (ALT and AST) activity. As shown in Figure 3A, liver cells were about the same size and orderly arrangement in the hepatic lobule of normal mice. Compared with the regular hepatic cellular pattern of the control group, the morphology of liver in the OVA challenged mice showed characteristic features such as slightly edematous cells, loose cytoplasm, and the bile duct hyperplasia in the local portal area. Obvious improvements in hepatic pathologic changes were observed in the middle and high dose RA intervention groups. As for the activities of transaminase, OVA challenge remarkably (*p <* 0.01) increased the serum AST and ALT activities (model group, AST: 185.7 U/L; ATL: 64.0 U/L) compared to the control mice (control group, AST: 49.2 U/L; ATL: 27.0 U/L). However, RA treatment notably (*p* < 0.01) attenuated the OVA challenge-induced elevation of serum ALT and AST activities (Figure 3B,C). These results indicate that RA could protect the liver from OVA challenge-induced liver dysregulation.

### 2.4. Effects of RA on OVA Challenge-Induced Liver Oxidative Stress 

The oxidative stress induced by OVA challenge in the liver was evaluated by assessing the concentration levels of MDA, NO, GSH, and the activity of SOD. As shown in Figure 4A, the concentration of MDA, a major product of lipid peroxidation, was remarkably (*p <* 0.01) increased in OVA challenged mice compared with the control group (control: 4.3 nmol/gprot; model: 11.2 nmol/gprot) while 14 consecutive days of RA intervention significantly (*p* < 0.01) decreased the MDA concentration compared to the RA allergic mice in a dose dependent manner (low: 6.8 nmol/gprot; middle: 6.1 nmol/gprot; high: 4.3 nmol/gprot). For NO, the pattern of the concentration change was consistent with MDA. RA treatment dose-dependently decreased the elevated NO concentration caused by OVA challenge (control: 0.8 μmol/gprot; model: 2.4 μmol/gprot; low: 1.6 μmol/gprot; middle: 0.8 μmol/gprot; high: 0.7 μmol/gprot.) (Figure 4B). GSH is a critical intracellular antioxidant that plays a central role in the reduction of hydroperoxides. The effects of RA on GSH levels are shown in Figure 4C. OVA challenge led to a significant (*p* < 0.01) reduction in the GSH level when compared to the normal group (control: 5.2 μmol/gprot; model: 3.0 μmol/gprot). However, the GSH levels were remarkably increased by treatment with low, middle, and high doses of RA (low: 4.0 μmol/gprot; middle: 4.4 μmol/gprot; high: 4.6 μmol/gprot). RA intervention significantly (*p* < 0.01) inhibited the decrease in the GSH level in the liver of OVA-allergic mice. SOD, a specific antioxidant enzyme scavenging O_2_^-^, acts as the first antioxidant defense line. The SOD levels dropped significantly (*p* < 0.01) in the allergic mice without RA intervention (control: 18.1 U/mgprot; model: 11.8 U/mgprot). The RA treatment dose dependently alleviated the decrease in SOD activity caused by OVA challenge (low: 13.5 U/mgprot; middle: 15.9 U/mgprot; high: 16.3 U/mgprot) (Figure 4D). These results, taken together, showed that RA intervention could relieve oxidative stress through balancing the OVA challenge-induced changes of the oxidative stress parameters and activity of antioxidant enzymes. 

### 2.5. Effects of RA on Cytokine mRNA Expression in Liver

To find out how RA affected the liver inflammation of OVA-induced allergic mice, the mRNA expression of inflammatory cytokines (TNF-α, IL-6, IL-4, TGF-β, MCP-1, iNOS, IL-1β, IL-10, and FOXP-3) were evaluated by quantitative real-time PCR. The results showed that the mRNA expression of typical inflammatory factors (including TNF-α, IL-6, IL-4, TGF-β, MCP-1, and iNOS) were notably (*p <* 0.01) increased in the liver of OVA-induced allergic mice. RA intervention remarkably (*p <* 0.01) alleviated the upregulation of these inflammatory factors (Figure 5A–F). As for the anti-inflammatory factors (IL-10 and FOXP-3), OVA challenge notably (*p <* 0.01) inhibited their expression while the dropped expression of IL-10 and FOXP-3 remarkably (*p <* 0.01) recovered in the liver of the RA-treated mice (Figure 5H,I). These results indicate that the regulation of inflammatory factor expression plays a crucial part in alleviating food allergy-induced liver injury by RA.

## 3. Discussion

RA, an important active ingredient of various medical herbs, possess remarkable biological effects including anti-inflammatory, hepatoprotective, cardioprotective, and anti-allergic activities [17,19]. In terms of anti-anaphylaxis activity, RA has been widely reported to exert pharmacological effects on atopic dermatitis, asthma, and allergic rhinitis [20]. However, its antiallergic bioactivity has been seldomly reported on intestinal food allergy. In this study, an OVA-induced intestinal food allergy mice model was established to evaluate the antianaphylaxis and hepaprotective activities of RA. RA intervention observably alleviated OVA-induced food allergy and liver dysregulation. Based on findings from the analysis of inflammatory cytokines and oxidative stress related indicators, the inhibition of oxidative damage and inflammation played vital roles in the hepaprotective activity of RA on OVA-induced intestinal food allergy.

Food allergy induced by OVA can result in a series of anaphylaxis symptoms including scratching, trembling, diarrhea, convulsions, and hypothermia. Many natural derived immunomodulating components such as flavonoid, polysaccharide, polyphenols, phycocyanin, and probiotics have been widely reported to exert anti-allergic activity on OVA-induced intestinal food allergy [21,22]. The immunomodulatory impacts of RA also make it a novel potential agent for the treatment of immune disorders of the body [23]. In allergic rhinitis, dermatitis, and asthma, RA showed obvious beneficial effects in alleviating allergic reactions such as mild anaphylactic symptoms, reduction in specific IgE and immune-mediators, and relieved tissue damage [24,25,26]. In this study, RA intervention at 30, 90, and 270 mg/kg significantly inhibited the OVA challenge-induced diarrheal, anaphylactic symptoms, and hypothermia (Figure 1B,C). Moreover, consistent with the effect in rhinitis, dermatitis, and asthma, RA also remarkably inhibited the secretion of OVA-sIgE, histamine, and mMCP-1 in the serum (Figure 2). Like all of the other widely reported natural polyphenols such as quercetin, baicalein, and epicatechin [27,28], RA shows great potential and application value in the relief of food allergy. Further studies of the related mechanisms are undoubtedly needed before any prophylactic and therapeutic use.

Generally, the liver is an immune tolerant organ that is uniquely equipped with a special immune structure composition to limit hypersensitivity to certain food antigens [29] whereas prolonged exposure to antigens in the liver may lead to imbalance of the gut–liver axis, which in turn mediates the liver immune response, and ultimately causes liver damage [9]. Allergy induced-gastroenteropathy complicated with hepatic dysfunction has been commonly reported in cases of food allergy [9,30,31]. The activities of serum ALT and AST are indicators of hepatotoxicity [32]. In our study, significant increases in ALT and AST activities were observed in the serum of OVA-challenged mice compared to the mice in the control group; however, these increases were notably attenuated by RA intervention (Figure 3B,C). Meanwhile, H&E staining also demonstrated a similar conclusion (Figure 3A). These suggest that RA has the great potential to alleviate liver injury mediated by food allergens. In allergy mediated rhinitis, dermatitis, and asthma, oxidative stress-induced liver injury has been extensively studied and reported [33,34,35]. Indeed, in this study, hypersensitivity in the form of oxidative stress in the liver was also observed in the OVA-allergic mice. This anaphylaxis was manifested by the high levels of MDA and NO, and the remarkable decrease in SOD activity and GHS levels in the liver (Figure 4). These changes in the hepatic oxidative indices were in agreement with those in the OVA-induced liver hypersensitivity reported before [36]. Polyphenol possesses a wide range of antioxidant activities. Many natural polyphenols such as ferulic acid, ECGC, and resveratrol have been reported to exert hepatoprotective activity through antioxidant activity [37,38,39]. Hereon, RA has also been shown to inhibit the increase of MDA and NO, the decrease in GSH, and reduced activity of SOD (Figure 4). These imply that the regulation of oxidative stress may play a vital role in the protectivity of RA in liver injury induced by food allergy.

Hepatic immune regulations are necessary to maintain liver homeostasis and would lead to liver pathology and dysfunction, if the homeostasis of the liver is disrupted [40]. TNF-α is directly toxic to hepatocytes via induction of cell apoptosis and mediates the expression of iNOS, which via NO plays a critical role in liver damage [41,42]. In our study, the prevention of TNF-α expression by RA intervention in turn prevented iNOS expression and the release of NO in the liver (Figure 5A,F and Figure 4B). IL-6 and IL4, which play active roles in host immune response, are critical inducers of cytokine storm in liver injury [43,44]. The elevated expression of IL-4 and IL-6 in the liver of OVA-allergic mice dramatically dropped after RA intervention (Figure 5B,C). FOXP-3, the main regulator of regulatory T (Treg) cell activity, could control the immune-suppressive response of regulatory T cells [45]. Meanwhile, IL-10, a cytokine with anti-inflammatory properties, plays a central role in infection by limiting the immune response and preventing damage to the host. In our research, downregulated expression of IL-10 and FOXP-3 in the liver of allergic mice was notably inhibited by RA intervention (Figure 5H,I). All of the cytokines expression analyses imply that the inhibition effect of RA on the elevation of pro-inflammatory cytokines and decline in anti-inflammatory cytokines plays a vital role in the hepatoprotective effect of RA on food allergy.

## 4. Materials and Methods

### 4.1. Materials and Reagents

RA, OVA, and aluminum hydroxide were from Thermo Fisher Scientific (Waltham, MA, USA). The Bicinchoninic Acid (BCA) Protein Quantitative Kit was provided by Beyotime Biotech (Shanghai, China). ELISA kits for OVA-sIgE, histamine, and mMCP-1 were from Fine Test Biotech (Wuhan, China). ALT, AST, MDA, SOD, GSH, and NO assay kits were purchased from Nanjing Jiancheng (Nanjing, China). TRIzol reagent for the RNA Extraction and cDNA Synthesis Kit were provided by Vazyme Biotech (Nanjing, China). 

### 4.2. Animal 

BALB/C mice (female, 4–6 weeks old, 16–18 g) were obtained from the SPF (Beijing) Biotechnology Co. Ltd. (Beijing, China). The health status of mice was verified by professional organizations (Suzhou Xishan biotechnology INC.). Mice were kept in a pathogen free environment maintained at 22 ± 2 °C and acclimatized for 7 days before the experiments. Food and water were provided ad libitum. All of the experiments were approved by the Animal Care and Use Committee of Huazhong University of Science and Technology (Wuhan, China; authorization number: SYXK (e) 2021-0057).

### 4.3. Animal Experimental Design

After adaptive feeding for 7 days, the mice were randomly divided into the five following groups (eight mice per group) according to their body weight: (1) the phosphate buffered saline (PBS) treated group (i.e., control); (2) OVA-challenged group (i.e., model); (3) RA (30 mg.kg^−1^ body weight) intervention group (i.e., RA-Low); (4) RA (90 mg.kg^−1^ body weight) intervention group (i.e., RA-Middle); (5) RA (270 mg.kg^−1^ body weight) intervention group (i.e., RA-High). In the model and RA intervention groups, mice were sensitized with OVA (5 mg.kg^−1^ body weight) with 2 mg of aluminum hydroxide in 200 μL PBS buffer through intraperitoneal injection on days 0 and 14. RA (30 mg.kg^−1^, 90 mg.kg^−1^, and 270 mg.kg^−1^ body weight) was administered via intragastric administration once a day to OVA sensitized mice from days 27 to 40. From day 28, mice (model group and RA intervention groups) were orally challenged with OVA (2.5 g.kg^−1^ body weight) in PBS every third day, for five times. On the OVA challenge days, the mice were deprived of food and water 1 h before the challenge. Sham-sensitized mice with PBS were the controls. After the last challenge, the anaphylactic symptoms, diarrhea rates, and rectal temperature were evaluated (Figure 1A). The allergic symptoms were scored according to the standards listed as Table 1. Rectal temperatures were measured by a rectal electronic thermometer.

### 4.4. Measurements of Specific Antibodies and Allergic Mediators in Serum

At one hour after the last challenge, serum from the tail vein blood was collected and stored at −80 °C prior to the evaluation of histamine and mMCP-1 using the commercial ELISA Kit from FineTest (Wuhan, China). Furthermore, all mice were sacrificed on day 41, and the levels of serum OVA-sIgE were measured using an anti-OVA-sIgE ELISA Kit from FineTest (Wuhan, China). 

### 4.5. Histopathologic Examination

After sacrifice, the livers of mice were collected and soaked in 10% neutral formalin buffer. After fixation, tissues were subsequently embedded in paraffin. The tissue sections (5 µm) were affined on the slides and stained with hematoxylin and eosin (H&E). Finally, the tissues were examined using a light microscope (Olympus, Tokyo, Japan) in random order without knowledge of the animal or group.

### 4.6. Analysis of Serum and Liver Biochemical Parameters

The activities of AST and ALT in the serum were measured according to the instructions. Liver tissues were homogenized in PBS at 4 °C. The liver homogenates were centrifuged at 3000× *g* for 10 min at 4 °C. Then, the supernatant after centrifugation was collected to determine the levels of MDA, GSH, and NO, and the activity of SOD using corresponding commercial kits (Nanjing Jiancheng Technology Co. Ltd., Nanjing, China). The protein content was determined with a BCA Kit.

### 4.7. RNA Preparation and Real-Time PCR

TRIzol reagent was used to extract the total RNA from liver tissue (10–30 mg). The quality and amount of the RNA were determined using a NanoDrop 2000 spectrophotometer. Reverse transcription was conducted with a cDNA Synthesis Kit. The primer sequences used are listed in Table 2. The quantitative real-time PCR was performed using SYBR Green Master on a Bio-Rad CFX96 system. The conditions for q-PCR referred to those we reported earlier [46]. The primer concentrations and Tm of these genes are listed in Table A1. The basic conditions are listed as follows: initial desaturated at 95 °C for 2 min, followed by 40 cycles of denaturation at 95 °C for 10 s, and annealing at 60 °C for 30 s. GAPDH was used for the normalization of the genes. The mRNA expression levels were assessed using the 2^−ΔΔCT^ method.

### 4.8. Statistical Analysis

All of the data are presented as the mean ± standard deviations (SD). We checked the data for Gaussian distribution by the Kolmogorov–Smirnov test. The statistical significance of the data with Gaussian distribution were determined by one-way analysis of variance (ANOVA) and Tukey’s test while the other data were analyzed using the non-parametric test. GraphPad Prism 9.0 was used to analyze the data. Results were considered statistically significant at *p* < 0.05.

## 5. Conclusions

RA, an active polyphenol found in many medical herbs, could significantly alleviate hypersensitivity symptoms and reduce mast cell protease levels, the histamine level, and specific IgE antibody levels in allergic mice. The results gathered from histological and transaminase activity analysis showed that RA reduced the degree of pathological changes in the liver of OVA-allergic mice. Homeostasis regulation of RA on oxidative stress and inflammation in the liver of OVA-allergic mice played a vital role in its hepatoprotective activity. These results can provide a theoretical basis for understanding the pathways by which RA exert hepatoprotective activity in a food allergic condition. Future studies will aim to elucidate the underling mechanism at the signal transduction level.

## Figures and Tables

**Figure 1 molecules-28-00788-f001:**
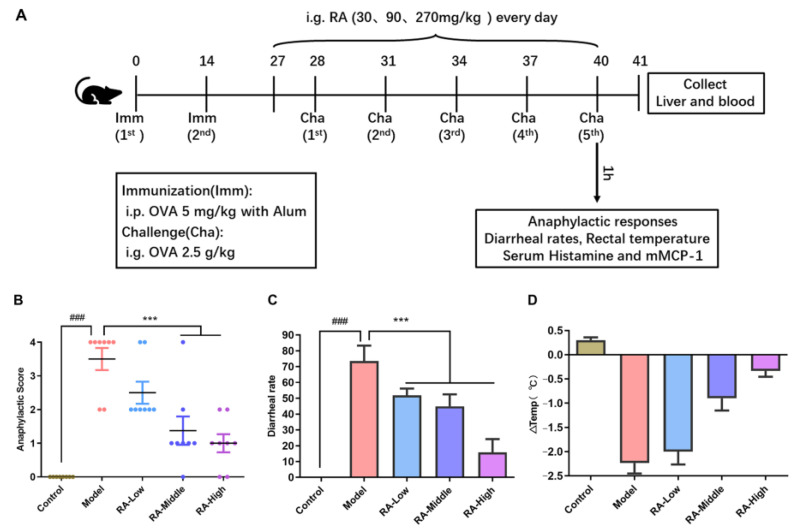
Procedure of animal experiments on mice and the effects of rosmarinic acid on the allergy symptoms. (**A**) Procedure for the mice food allergy model. Groups of mice (eight mice per group) were sensitized twice by intraperitoneal injection of OVA (5 mg/kg body weight) on days 0 and 14. From days 28 to 40, the mice were intragastrically administered with OVA (2.5 g.kg^−1^ body weight) five times. Meanwhile, mice in the RA intervention group were orally challenged with RA (30, 90, and 270 g.kg^−1^ body weight) once every day from days 27 to 40. One hour after the last challenge, anaphylaxis symptoms and related anaphylaxis serum indicators were evaluated. (**B**) Score of anaphylactic symptoms at 1 h after the last challenge. (**C**) Diarrhea rates at 1 h after the last challenge. (**D**) Rectal temperature differences at 45 min after the last challenge. ^###^
*p* < 0.01, compared with the control group; *** *p* < 0.01 compared with the OVA-allergic model group.

**Figure 2 molecules-28-00788-f002:**
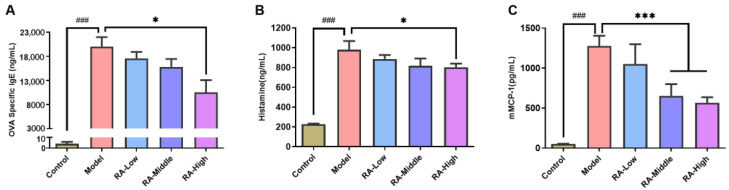
Effects of RA on OVA-sIgE and anaphylactic mediators of OVA-sensitized mice. (**A**) Effects of RA on OVA-sIgE. At day 41, the serum from eyeball blood was collected and the OVA-sIgE was measured using an ELISA Kit. (**B**) Effects of RA on the histamine levels. Serum from the caudal vein of mice was collected at 1 h after the last challenge. The histamine levels were measured using ELISA. (**C**) Effects of RA on mMCP-1 levels. The mMCP-1 levels were measured by ELISA. *^###^ p* < 0.01, compared with the PBS group; * *p* < 0.1, *** *p* < 0.01 compared with the OVA model group.

**Figure 3 molecules-28-00788-f003:**
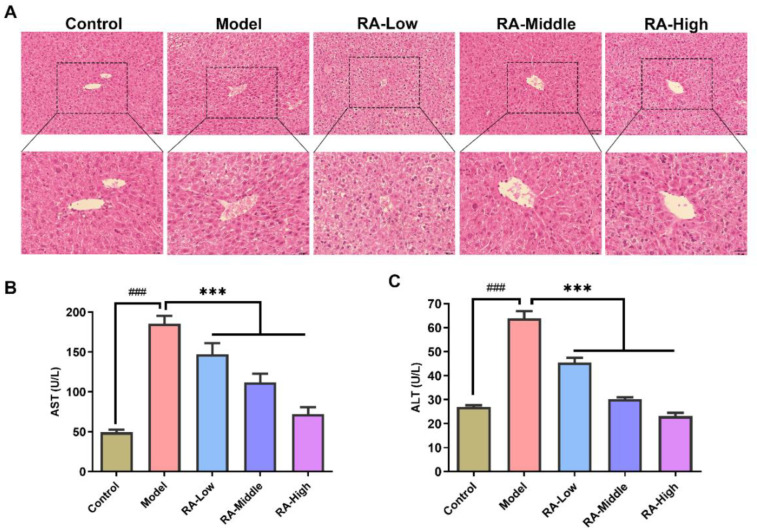
Effects of RA on the serum ALT, AST levels, and liver histopathological features OVA-allergic mice. (**A**) H&E staining analysis of the histological sections of liver. The upper panel: Scale bar = 100 μm. The lower panel: Scale bar = 50 μm. (**B**) Effect of RA on serum AST activity. At day 41, the serum from the caudal vein blood was collected and the activity of AST was measured by the assay kit. (**C**) Effect of RA on the serum ALT activity. ^###^
*p* < 0.01 compared with the PBS group; *** *p* < 0.01 compared with the OVA-allergic model group.

**Figure 4 molecules-28-00788-f004:**
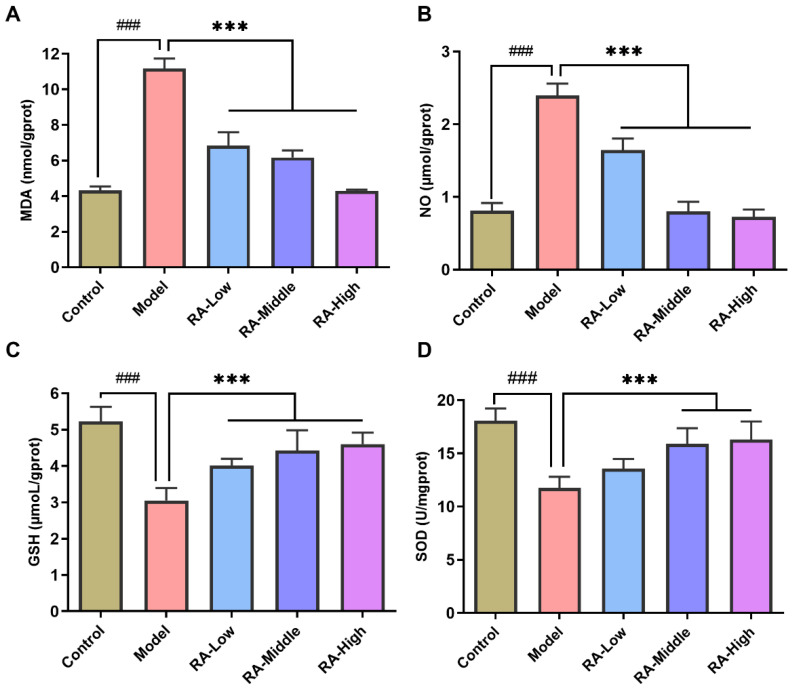
Effects of RA on the liver MDA, NO, SOD, and GSH levels. Liver levels of MDA (**A**), NO (**B**), GSH (**C**), and activity of SOD (**D**) were assayed. ^###^
*p* < 0.01 compared with the PBS group; *** *p* < 0.01 compared with the OVA model group.

**Figure 5 molecules-28-00788-f005:**
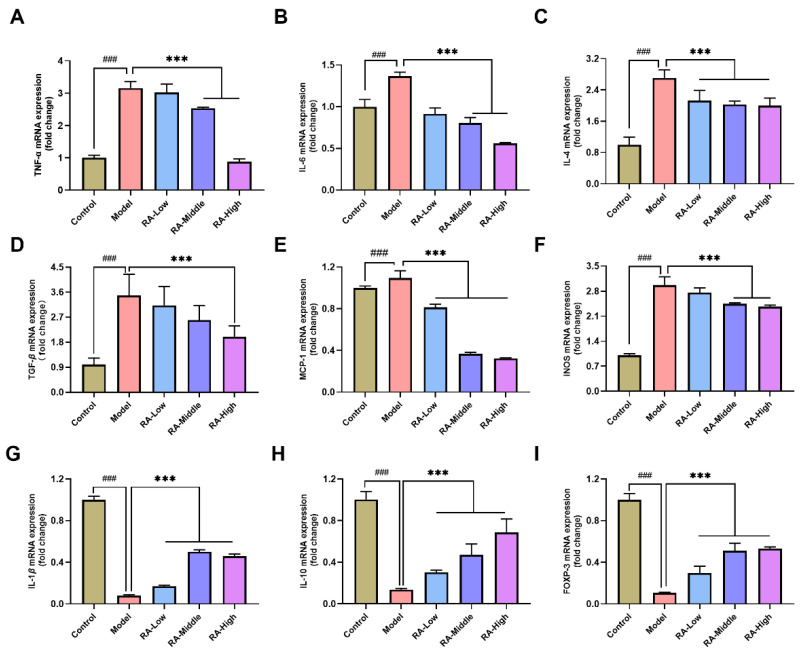
Effects of RA on the expression of cytokines on the liver of OVA-induced allergic mice. The mRNA expression fold changes of cytokines (TNF-α (**A**), IL-6 (**B**), IL-4 (**C**), TGF-β (**D**), MCP-1 (**E**), iNOS (**F**), IL-1β (**G**), IL-10 (**H**) and FOXP-3 (**I**)) were analyzed by real-time fluorescent quantitative PCR. ^###^
*p* < 0.01 compared with the PBS group; *** *p* < 0.01 compared with the OVA-allergic model group.

**Table 1 molecules-28-00788-t001:** Allergy symptoms.

**Scores**	**Allergy Symptoms**
0	Without any symptoms
1	Scratching and rubbing around the mouth and nose
2	Swelling around the mouth and eyes; diarrhea; reduced activity; increased respiratory
3	Wheezing; labored respiration; cyanosis around the mouth and the tail
4	No activity after stimulation; shivering; muscle contractions
5	Death by shock

**Table 2 molecules-28-00788-t002:** Primer sequences for quantitative real-time PCR.

Genes	Forward Primer	Reverse Primer	Product Size (bp)
GAPDH	AGGTCGGTGTGAACGGATTTG	TGTAGACCATGTAGTTGAGGTCA	123
TNF-α	CCCTCACACTCAGATCATCTTCT	GCTACGACGTGGGCTACAG	61
IL-6	TAGTCCTTCCTACCCCAATTTCC	TTGGTCCTTAGCCACTCCTTC	76
IL-4	GGTCTCAACCCCCAGCTAGT	GCCGATGATCTCTCTCAAGTGAT	102
IL-1β	GCAACTGTTCCTGAACTCAACT	ATCTTTTGGGGTCCGTCAACT	89
MCP-1	GAGGACAGATGTGGTGGGTTT	AGGAGTCAACTCAGCTTTCTCTT	231
TGF-β	CTCCCGTGGCTTCTAGTGC	GCCTTAGTTTGGACAGGATCTG	133
iNOS	GTTCTCAGCCCAACAATACAAGA	GTGGACGGGTCGATGTCAC	127
IL-10	GCTCTTACTGACTGGCATGAG	CGCAGCTCTAGGAGCATGTG	105
FOXP-3	CCCATCCCCAGGAGTCTTG	ACCATGACTAGGGGCACTGTA	183

## Data Availability

The data presented in this study are available on request from the corresponding author.

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
