# Peer review of "Hepatoprotective Effects of Rosmarinic Acid on Ovalbumin-Induced Intestinal Food Allergy Mouse Model"

_molecules, 2023, doi:10.3390/molecules28020788_

Round 1

Reviewer 1 Report

The title as well as keywords accurately reflects the major findings of the work.

The abstract adequately summarize methodology, results, and significance of the study. Authors should simplify the introductory paragraph and better describe the aim of the study.

Authors should indicate statistical analysis applied together with the results (i.e. P value).

The introduction section falls within the topic of the study, however, Authors should enhance this section adding more information concerning the diet supplementation in veterinary field emphasizing the significant increase of interest showed by scientific community on diet improvement to enhance animal health status and welfare.  On this regards, after the two paragraphs of introduction and before the sentence “Rosmarinic acid (RA), an ester of caffeic acid and 3,4-dihydroxyphenyl lactic acid, is an active polyphenol found in medical herbs, including Ocimum basilicum, Salvia miltiorrhiza Bunge, Rosmarinus officinalis and Origanum vulgare [13].” Authors could add the following information and the related references “Nowadays, the improvement of diet composition become a key factor to improve the health status and welfare of animals (Giannetto C. et al., Antioxidants 11, 2022, 2339; Abbate J.M. et al., Animals 2020, 10, 2303; Avondo M. et al., Journal of Dairy Research, 2009, 76: 202-209; Monteverde V. et al., Journal of Applied Animal Research, 2017, 45: 615-618).

The section of Materials and Methods is clear for the reader and it meticulously describes the methods applied in the study. However, Authors should check this section and correct many punctuation errors as well as English language. Moreover, some missing information should be added:

-Please indicate how the health status of animals has been verified.

-Authors should indicate the criteria chosen to divide in a homogeneous way the animals among groups.

-Regarding statistical analysis, did Authors checked the data for Gaussian distribution by Normality test?

Results section as well as Discussion section is clear and well written. The findings obtained in the study were well discussed and justified with appropriate references.

Authors well summarized the results and the significance of the study in the conclusion section, however, I suggest to change “Pathological results and transaminase activity analysis showed that they significantly reduced…” with “The results gathered from histological and transaminase activity analysis showed that RA reduced…”. 

The tables are generally good and well represent the results of the study. I suggest to improve the quality of figures.

Authors should check and standardize the references in the list according to journal guidelines.

Reviewer 2 Report

This study evaluates protective effects of rosmarinic acid on OVA-treated mouse model. Overall, the study is well designed. 

Major concerns:

1. Lack of validation of real-time pcr conditions. Have the authors previously optimized these conditions? If so, please cite that article or otherwise showed in supplementary table.  Some information that should be put in the supplementary table include primer conc., Ta, %efficiency, and R^2.

2. It is a rule of thumb that at least two reference genes must be used in real-time pcr.  Using only one reference gene has not been acceptable for several past years. If the authors have kept cDNA, I suggest you run another reference gene and use geometric average of GAPDH+new ref gene for calibration of target gene expression.

Minor concerns:

1. Abstract : edit FXOP-3 to FOXP-3

2. Page 1, Line 2 of Introduction : edit involving to involve

3. Page 2, Line 2 of second paragraph : change medical to medicinal

4. Page 2, section 2.1 : delete full term of ova since it was already present in introduction section.

5. Page 2, section 2.1 : spell out BCA

6. Page 2, section 2.1 : ...., GSH, NO assay kits were.....

7. Page 2, section 2.2 : change al libitum to ad libitum

8. Page 2, section 2.2 : Provide approval no. of animal use

9. Page 3, section 2.5 : Move full terms of AST and ALT to introduction section; and remove full terms of MDA, GSH, NO, and SOD

Reviewer 3 Report

In this review, the protective effects of rosmarinic acid on the liver and ovalbumin-induced intestinal food allergy were investigated in mice. The aim is fine. However, the study design is not appropriate. The cellular and molecular mechanisms are not investigated. Due to the well-known effect of rosmarinic acid on inflammation (PMID: 32184728). This study only briefly studies the anti-inflammatory effects were only studied in mRNA levels.

FXOP-3 is not correct in the abstract.

Change the color of the dots in Figure 1B.

The control bars can not be shown in Figure 2A.

The bar scale in Figure 3A is not correct.

Figure 5, the control should be balanced to 1 for calculating the fold-change of mRNA expression.

The abbreviation should be defined as the first time used in the abstract, as well as the main text. Some such as rosmarinic acid (RA), an ester of caffeic acid and 3,4-dihydroxyphenyl lactic acid, was defined in both the introduction and discussion.

Round 2

Reviewer 3 Report

The manuscript was improved with revisions. No further comment.